# Cricket Meal (*Gryllus bimaculatus*) as a Protein Supplement on In Vitro Fermentation Characteristics and Methane Mitigation

**DOI:** 10.3390/insects13020129

**Published:** 2022-01-25

**Authors:** Burarat Phesatcha, Kampanat Phesatcha, Bounnaxay Viennaxay, Maharach Matra, Pajaree Totakul, Metha Wanapat

**Affiliations:** 1Department of Agricultural Technology and Environment, Faculty of Sciences and Liberal Arts, Rajamangala University of Technology Isan, Nakhon Ratchasima 30000, Thailand; Burarat_kat@hotmail.co.th; 2Department of Animal Science, Faculty of Agriculture and Technology, Nakhon Phanom University, Nakhon Phanom 48000, Thailand; kamphe@npu.ac.th or; 3Tropical Feed Resources Research and Development Center (TROFREC), Department of Animal Science, Faculty of Agriculture, Khon Kaen University, Khon Kaen 40002, Thailand; bounnaxay@gmail.com (B.V.); mos-matra@hotmail.com (M.M.); pajaree_totakul@hotmail.com (P.T.)

**Keywords:** cricket meal, protein, ruminal fermentation, digestibility

## Abstract

**Simple Summary:**

Protein sources of high quality and sustainability are found in insects. In many regions, insects are a primary food source, such as in Africa, South America, Asia, and Oceania. Insects are considered promising alternative feed sources, in particular as a source of protein. The use of edible insects as high-protein sources is widespread, and cricket has been proved to be a potential food and feed insect species. Cricket (*Gryllus bimaculatus*) also contain 54.10% crude protein, 6.90% crude fiber, 26.90% fat, and 78.90% total digestible nutrient, as well as a variety of essential amino acids, including methionine, lysine, histidine, valine, and leucine. In addition, insects have been investigated as a source of protein in diets of poultry, swine, and fish. However, there are currently little data on the utilization of insects as ruminant feed. The objective of this experiment was to conduct the effects of Cricket meal (*Gryllus bimaculatus*) (CM) as a protein replacement for soybean meal on in vitro fermentation end products, gas production, nutrient degradability, and methane mitigation.

**Abstract:**

The aim of this work was to conduct the effects of cricket (*Gryllus bimaculatus*) meal (CM) as a protein supplement on in vitro gas production, rumen fermentation, and methane (CH_4_) mitigation. Dietary treatments were randomly assigned using a completely randomized design (CRD) with a 2 × 5 factorial arrangement. The first factor was two ratios of roughage to concentrate (R:C at 60:40 and 40:60), and the second factor was the level of CM to replace soybean meal (SBM) in a concentrate ratio at 100:0, 75:25, 50:50, 25:75, and 0:100, respectively. It was found that in vitro DM degradability and the concentration of propionic (C_3_) were significantly increased (*p* < 0.05), while the potential extent of gas production (a + b), acetate (C_2_), acetate and propionate (C_2_:C_3_) ratio, and protozoal population were reduced (*p* < 0.05) by lowering the R:C ratio and the replacement of SBM by CM. In addition, rumen CH_4_ production was mitigated (*p* < 0.05) with increasing levels of CM for SBM. In this study, CM has the potential to improve rumen fermentation by enhancing C_3_ concentration and DM degradability, reduced methane production, and C_2_:C_3_ ratio. The effects were more pronounced (*p* < 0.05) at low levels of roughage.

## 1. Introduction

The demand for animal products such as meat, milk, and eggs has risen dramatically in recent years as a result of the rapidly expanding human population. A promising alternative to traditional livestock to meet rising demand for meat-like products is insects. Insect protein has advantages over plant and animal protein sources, including rapid conversion from organic substrate to insect biomass, low water consumption, and low greenhouse gas emissions [1]. Greenhouse gas emissions from livestock farming are high, mainly caused by methane emanating from rumen fermentation [2]. Various dietary and animal management techniques have been proposed to reduce ruminant methane emissions [3]. Feeding is a significant cost factor in livestock production, and feed costs contribute 60–80 percent of overall production expenses [4]. Hence, it is interesting to search for alternative feed opportunities as a strategy for meeting future feed needs. One strategy from the global community is to investigate new protein sources. In Africa, South America, Asia, and Oceania, insects have been a traditional food source in these regions but have recently also gained interest as alternative protein sources in other parts such as Europe and North America [5]. Feeding insects to both non-ruminants and ruminants has recently been recommended [6]. Several insect species have been evaluated for their potential as feed for domesticated animals, including black soldier fly larva, mealworm, grasshopper, locust, house fly maggot, silkworm, and cricket [5]. Worms are readily available, easily accessible, and widely accepted as a source of high-quality protein (55%) in the form of food or feed [7]. Marareni and Mnisi [8] reported that mopane worm (*Imbrasia belina*) showed potential to replace soybean meal in Jumbo quail feed without compromising their performance and health status. Recently, the use of edible insects as high-protein feed sources has increased, and cricket has shown economic production potential [9]. Cricket (*Gryllus bimaculatus*) contains 54.10% crude protein (CP), 6.90% crude fiber (CF), 26.90% fat, and 78.90% total digestible nutrient (TDN) (DM basis), as well as a variety of essential amino acids including methionine, lysine, histidine, valine, and leucine [5]. Wang et al. [10] reported that cricket also contained 8.7% chitin. Chitin or chitosan supplementation enhanced chitosan glucosamine interaction with bacterial cell walls, and the resulting permeability changes reduced the population of bacteria [11]. Frye and Calvert [12] found that silkworm larvae had low chitin content and were easier to digest than other insects commonly used as food sources. Their data revealed chitin content of silkworm larvae similar to crickets. Crickets are also considered an excellent source of other macro minerals such as Mg, Fe, Ca, K, and Na [13]. Insects have been investigated as a protein source in diets of poultry, swine, and fish [9,14]. Increasing the amount of *Hermetia illucens* meal in fish diet led to an increase in saturated fatty acids (SFAs), particularly lauric acid (12:0), with a decrease in valuable polyunsaturated fatty acids (PUFAs) [15,16]. Cullere et al. [17] revealed that broilers fed *Hermetia illucens* larva meal had higher levels of saturated fatty acids (SFAs), particularly lauric acid (12:0), and lower levels of valuable polyunsaturated fatty acids (PUFAs) in their meat. Microbial degradation and fermentation on the substrate and buffering of volatile fatty acid (VFA) production were the main sources of gas in in vitro rumen fermentation, according to Getachew et al. [18]. Oil or fat, regardless of source, reduced methane emissions, while increased dietary fat concentration and decreased DM and neutral detergent fiber (NDF) digestibility [19]. Oil inhibits methanogenesis by reducing the population or activity of methanogens, eliminating some protozoa, and decreasing nutrient degradation and fermentation. Field cricket (*Teleogryllus mitratus*), fed as a partial replacement for soybean meal in the diet of 8–20-day broilers, resulted in no negative effects on growth, weight gain, or feed intake [20], while Permatahati et al. [21] found that replacement of fish meal by cricket meal resulted in increased egg production and egg quality of layer quails. However, limited research is available on the use of insects as ruminant feed. The objective of this experiment was to evaluate the effect of cricket meal (*Gryllus bimaculatus*) (CM) as a protein replacement for soybean meal (SBM) on in vitro fermentation end-products, gas production, nutrient degradability, and methane mitigation.

## 2. Materials and Methods

This study was approved by the Animal Care and Use Committee of Khon Kaen University, Thailand. The experimental feeds were randomly assigned in a 2 × 5 factorial arrangement in a completely randomized design (CRD). Treatments were various levels of roughage to concentrate (R:C) ratio and replaced soybean meal (SBM) with cricket meal (CM). R:C in a ratio of 60:40 and 40:60, and the replaced SBM with CM in a concentrate ratio at 100:0, 75:25, 50:50, 25:75, and 0:100, respectively. Each treatment contained three runs and three replications (10 treatments × 3 replications + 3 bottles of blank).

This study used an adult cricket meal, obtained from cricket farm in Thailand. The cricket farming and the cricket flour production were from Khon Kaen, Thailand. Rice straw, concentrate, and cricket meal were dried by hot air oven at 60 °C and then ground to a length of 1 mm (Cyclotech Mill, Tecator, Hoganas, Sweden). Samples of various R:C ratios and concentrate mixtures were used for chemical analysis and in the in vitro gas production experiment. The samples were chemically analyzed by standard methods for dry matter (DM), ash, crude protein (CP), ether extract (EE) [22], neutral detergent fiber (NDF), and acid detergent fibre (ADF), according to Van Soest et al. [23]. This experiment used rumen fluid as a fermentation source from two rumen-fistulated dairy steers (75% Holstein Friesian and 25% Thai native breed, 3 years old) with an average live weight of 320 ± 10 kg. Before the morning feeding, about 1000 mL of rumen liquor was collected and combined from each animal. Dairy steers were fed with rice straw ad libitum and concentrate mixture (14% CP, 75% TDN) at 0.5% of body weight. The method used for in vitro fermentation technique was that of Menke et al. [24], as modified by Kang et al. [25]. During the incubation, gas production of all treatment samples was recorded at 1, 2, 4, 6, 8, 12, 24, 48, 72, and 96 h. The cumulative gas produced during fermentation was fitted into the model of Orskov and McDonald [26].

When the inoculum ruminal fluid was obtained at 4, 8, and 12 h after the inoculation, gas production was recorded at each time point, and pH was measured immediately using a portable pH temperature meter. Then, the samples of rumen fluid were prepared into two parts. The first portion of 20 mL was centrifuged at 16,000× *g* for 15 min and then the supernatant was taken and stored at −20 °C. Ammonia nitrogen (NH_3_-N) and a concentration of volatile fatty acids (VFAs), which was 25 mL of filtered fluid sample, was mixed with 2.25 mL of 1 M H_2_SO_4_, and then it was centrifuged at 16,000 g for 15 min, and the supernatant was used for NH_3_-N analysis using the micro-Kjeldahl method [22]. Volatile fatty acids (VFAs); (C_2_, C_3_, and C_4_) analysis was analyzed by HPLC [27]. The second portion, a total direct counting approach, was used on the protozoal population utilizing a hemocytometer, according to the method of Galyean [28]. After incubation, in vitro DM degradability (g/kg) was determined at 12 and 24 h [29]. Estimation of (CH_4_) production of rumen was estimated using the VFA proportions [30].

All the experimental data were statistically analyzed by the general linear model (GLM) according to SAS [31]. Treatment means were statistically performed using Tukey’s multiple comparison test [32]. Comparisons between the R:C ratio and CM replacement SBM were tested by Duncan’s new multiple range test. Differences among means with *p* < 0.05 were accepted as being statistically significant.

## 3. Results

### 3.1. Chemical Composition of Experimental Feeds

Chemical compositions of concentrate, cricket meal, and rice straw are listed in Table 1. Concentrate mixtures and CP concentrations were similar among the formulas, with CP ranging from 14.0 to 14.4% and CM and rice straw containing 68.5 and 2.2% CP, respectively. CM contained 94.2% DM, 3.4% ash, 33.1% NDF, 12.7% ADF, and 12.5% EE, while rice straw contained 91.6% DM, 9.3% ash, 75.5% NDF, and 47.4% ADF.

### 3.2. Measurement of Parameters Based on In Vitro Gas Production Technique and Rumen Degradability of Nutrients

Gas production from the immediately soluble fraction (a) was not affected (*p* > 0.05) by the R:C ratio, while SBM replacement with CM had no effect on fraction (b) and fraction (c) (Table 2). The potential extent of gas production (a + b) and cumulative gas production at 96 h were affected by the R:C ratio. Only 100% CM replacement had an effect (*p* < 0.05), and no interactions were observed. In vitro DM degradability (g/kg) at both 12 and 24 h was improved (*p* < 0.05) by increasing the R:C ratio and level of CM replacing SBM; however, no interactions were observed (*p* > 0.05).

Increasing levels of CM replacing SBM resulted in increased cumulative gas production at 96 h and in vitro DM degradability. Higher gas production and IVDMD were caused by higher concentrate proportion, resulting in higher NFC and lower NDF contents.

### 3.3. Volatile Fatty Acid and Methane Production

Data on total VFA, VFA proportions, and rumen CH_4_ production are presented in (Table 3 and Figure 1). All parameters except C_4_ were influenced by the R:C ratio, while increasing levels of CM replacing SBM significantly modulated (*p* < 0.01) C_2_, C_3_, C_2_:C_3_ ratio and rumen CH_4_ production. However, interactions of the R:C ratio and replacement of SBM by CM were not found in all parameters (*p* > 0.05). Decreasing the roughage to concentrate ratio increased DM degradability, total VFA, and C_3_ proportions, while decreasing C_2_ and CH_4_.

### 3.4. Ruminal pH, Ammonia–Nitrogen (NH_3_-N) Concentration and Protozoal Population

The pH of all treatments varied between 6.4 and 6.7, being lower for substrates with 40:60 R:C ratio compared to 60:40 (Table 4). The R:C ratio and replacement of SBM with CM was impacted at all levels, resulting in improved (*p* < 0.01) NH_3_-N concentration and protozoal population. The protozoal population reduced (*p* < 0.01) as a result of the R:C ratio, was significantly reduced (*p* < 0.05) in the 60:40 R:C ratio group, and increased in the 40:60 R:C ratio group, while interaction of the R:C ratio and CM replacing SBM had no effect on all parameters.

## 4. Discussion

Rice straw was found to be abundantly available in many countries and was fed ad libitum during the experimental period. Wanapat et al. [33] suggested that feeding rice straw with a concentrate mixture containing a high density of energy and protein could be beneficial in increasing its utilization. Insects, as a high protein source, have recently become a popular alternative option. Livestock diets can be supplemented with insects to increase protein concentration. Sánchez-Muros et al. [9] reported that cricket (*Gryllus bimaculatus*) was an excellent food and feed insect species. The cricket meal used in this experiment contained 68.5% CP, 33.1% NDF, 12.7% ADF, and 12.5% EE; this finding was similar to that of Jayanegara et al. [5], who stated that cricket meal contained 67.7% CP, 39.3% NDF, 10.8% ADF, and 14.5.% EE. Sánchez-Muros et al. [9] showed that insect ether extract (EE) contained polyunsaturated fatty acids (PUFAs). However, Chakravorty et al. [34] reported that cricket fatty acid profiles consist of 2.34% linoleic acid, 9.77% oleic acid, 32.06% stearic acid, and 50.32% palmitic acid. Furthermore, insects did not contain only high profiles of crude protein, essential amino acids, minerals, and unsaturated fatty acids (USFA), but also a variety of bioactive compounds such as lauric acid, chitin, peptidase, and flavonoids [35,36,37,38,39]. Inclusion of oil from insects in the diet reduced the production of methane. Lipids can reduce enteric CH_4_ levels by slowing the fermentation process.

Additionally, Jayanegara et al. [5] discovered that insect meals with a high fiber content had a lower IVDMD and IVOMD than SBM. The high fiber and EE content in insect meals also resulted in low total gas production. Getachew et al. [40] found that adding yellow grease and tallow to total mixed rations supplement results in decreased gas production and in vitro true digestibility. Dietary fat influences the particles of fiber in the rumen by covering the surface of fiber cells, thereby preventing degradation by microbes. Beauchemin et al. [41] noted that unsaturated fatty acids were not more detrimental to fiber digestion, while Jayanegara et al. [42] found that lower digestibility of insect meals results in lower levels of H_2_ production, which is a key substrate for methanogenesis. Oil or fat from a variety of sources reduced methane emissions. Patra [19] demonstrated that increasing fat content in the diet reduced methane emissions from cattle. Wu et al. [43] revealed that supplementing oleic acid in the diet reduced methane production, while increasing beneficial fatty acids in an in vitro study. In addition, the high EE content leads to reduced methane emissions because some fatty acids, particularly medium chain fatty acids (MCFA), are poisonous to methanogenic archea [44]. The variability in methane emissions is due to the level of insect oils supplementation in feed, suggesting that fatty acids in insect oils play a role in mitigating (CH_4_). In this investigation, rumen pH varied at 6.4–6.7, which is in agreement with Calabrò et al. [45], who reported that cellulolytic bacteria functioned normally at a pH of 6.3–7.0 in the rumen, while Russell and Rychlik [46], stated that ruminal pH is an important parameter that reflects the internal homeostasis of rumen ecology. Kazemi-Bonchenari et al. [47] demonstrated that higher amounts of protein in the diet lead to higher concentrations of ruminal NH_3_-N in the rumen. Proteins in the rumen are broken down into ammonia by proteolytic microbes [48]. Jayanegara et al. [49] showed that the concentration of rumen degradable protein in concentrate was higher than in forage, thereby contributing to higher ammonia production. In the present study, higher levels of concentrate diet and levels of CM used to replace SBM enhanced ruminal NH_3_-N concentration. This is most likely owing to the high protein content of CM. It could also be attributed to the ability of ruminal bacteria to produce stimulatory substances and even protein, or to changes in the abundance of microbes with proteolytic activity. Furthermore, changing the R:C ratio in the diet from 60:40 to 40:60 increased the ruminal NH_3_-N concentration. Similarly, Phesatcha et al. [50] and Viennasay et al. [51] discovered that rumen NH_3_-N concentrations increased dramatically with decreasing R:C ratio.

Protozoal populations tended to increase with the use of high-concentrate sources of carbohydrate, which benefited protozoal growth [52]. Boussaada et al. [53] reported that *Eucalyptus globulus* essential oil extracts decreased the methanogenic population and total number of protozoa in in vitro experiments due to the effect of secondary metabolite compounds in eucalyptus, which had antiprotozoal activities. Reducing the amount of H_2_ producers such as protozoa in the rumen is an essential strategy in lowering methane production [54]. Our results indicated that supplementation of CM containing oil mitigated protozoal numbers and CH_4_ production in the rumen. Ciliate protozoa play an important role in rumen methanogenesis due to their relationship with methanogens that attach to cell membrane surfaces [55]. Methanogenesis can be inhibited by the addition of oil that reduces the population or activity of methanogens by the partial elimination of protozoa. In addition, Belanche et al. [56] revealed that chitosan reduced ruminal methane emissions and methanogen populations.

## 5. Conclusions

Cricket meal is rich in protein and could be used to replace soybean meal as a protein supplement to improve propionate production in the rumen, reduce protozoal population, and mitigate methane production. However, further in vivo trials are required for a more detailed investigation of the use of CM as a source of protein.

## Figures and Tables

**Figure 1 insects-13-00129-f001:**
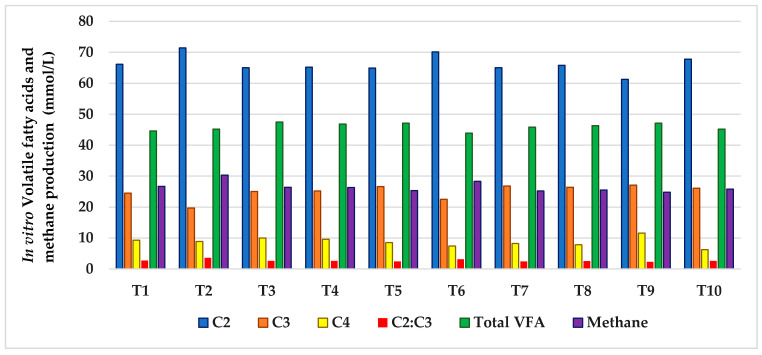
Effect of Cricket meal (*Gryllus bimaculatus*) (CM) on rumen volatile fatty acids (VFAs) and methane production. T1, 60:40 R:C + 0%CM; T2, 60:40 R:C + 25%CM; T3, 60:40 R:C + 50%CM; T4, 60:40 R:C + 75%CM; T5, 60:40 R:C + 100%CM; T6, 40:60 R:C + 0%CM; T7, 40:60 R:C + 25%CM; T8, 40:60 R:C + 50%CM; T9, 40:60 R:C + 75%CM; T10, 40:60 R:C + 100%CM.

**Table 1 insects-13-00129-t001:** Feed ingredients and chemical composition of concentrate, cricket meal (*Gryllus bimaculatus*), and roughage sources used in the experiment.

Items	Replacement Levels of CM for SBM (% Fresh Basis)					CM	RS	SBM
	0	25	50	75	100			
Ingredients (% as fed)								
Cassava chip	60.0	60.0	58.0	55.0	55.0			
Soybean meal	16.0	12.0	8.0	4.0	0.0			
Cricket meal	0.0	4.0	8.0	12.0	16.0			
Coconut meal	7.0	7.0	7.0	8.0	8.0			
Rice bran	7.0	7.0	8.0	7.8	7.6			
Palm meal	5.0	5.0	6.0	7.0	7.0			
Molasses	1.0	1.0	1.0	2.0	2.0			
Urea	1.0	1.0	1.0	1.2	1.4			
Sulfur	1.0	1.0	1.0	1.0	1.0			
Mineral mixed	1.0	1.0	1.0	1.0	1.0			
Salt	1.0	1.0	1.0	1.0	1.0			
Chemical composition								
Dry matter (%)	90.3	90.8	92.3	92.7	93.1	94.2	91.6	90.5
	------------------------------------% dry matter -----------------------------------							
Organic matter	93.6	94.5	94.1	93.6	94.4	96.6	90.7	94.3
Ash	6.4	5.5	5.9	6.4	5.6	3.4	9.3	5.7
Crude protein	14.0	14.1	14.0	14.3	14.4	68.5	2.2	40.1
Neutral detergent fiber	20.1	20.8	24.7	24.5	24.3	33.1	75.5	12.8
Acid detergent fiber	10.7	10.5	12.2	13.5	13.1	12.7	47.4	9.7
Ether extract	2.2	7.2	8.0	10.6	11.3	12.5	-	13.2

CM, cricket meal; SBM, soybean meal; RS, rice straw.

**Table 2 insects-13-00129-t002:** Effect of cricket meal (*Gryllus bimaculatus*) (CM) on gas production kinetics and in vitro DM degradability.

Treatments	R:C Ratio	SBM:CM	Gas Production Kinetics				Cumulative Gas Production at 96 h (mL/0.5 g DM Substrate)	In Vitro DM Degradability (%)	
			a	b	c	a + b		12 h	24 h
T1	60:40	100:0	−0.4	70.7	0.04	70.3	65.7	52.7	59.3
T2		75:25	−0.8	71.5	0.03	70.7	68.8	55.9	59.7
T3		50:50	1.3	69.2	0.04	71.5	69.7	57.0	60.2
T4		25:75	1.1	69.8	0.04	70.9	69.1	57.5	60.5
T5		0:100	1.6	69.1	0.04	70.7	70.2	56.5	61.6
T6	40:60	100:0	−3.1	76.4	0.03	73.3	65.3	62.8	62.8
T7		75:25	−2.7	74.3	0.03	66.1	66.7	66.5	62.4
T8		50:50	−0.8	71.5	0.03	70.7	68.7	64.6	64.0
T9		25:75	−2.2	73.4	0.03	71.7	70.1	65.1	64.1
T10		0:100	−1.1	70.8	0.05	69.7	72.1	69.6	65.6
SEM			0.38	0.58	0.008	1.42	1.18	0.45	0.58
Comparison R:C ratio SBM:CM R:C ratio × SBM:CM									
			0.06	0.03	0.001	0.001	0.001	0.001	0.001
			0.92	0.95	0.006	0.04	0.03	0.04	0.03
			0.84	0.85	0.12	0.14	0.78	0.36	0.28

R:C, ratio of roughage to concentrate at 60:40 and 40:60; SBM:CM, soybean meal replacement by cricket meal in a concentrate ratio at 100:0, 75:25, 50:50, 25:75, and 0:100, respectively; a, gas production from immediately soluble fraction, b, gas production from insoluble fraction, c, gas production rate constant for insoluble fraction, a + b, potential extent of gas; SEM, standard error of the mean.

**Table 3 insects-13-00129-t003:** Effect of cricket meal (*Gryllus bimaculatus*) (CM) on in vitro gas production, proportions of volatile fatty acids, and methane production.

Treatments	R:C Ratio	SBM:CM	Molar Proportions of VFA (mmol/L)				Total VFA (mmol/L)	CH_4_ (mmol/L)
			C_2_	C_3_	C_4_	C_2_:C_3_		
T1	60:40	100:0	72.0	19.0	8.9	3.8	59.8	30.7
T2		75:25	71.9	19.2	8.9	3.7	52.5	30.6
T3		50:50	71.5	20.5	7.9	3.5	56.3	29.7
T4		25:75	69.1	22.1	8.8	3.1	68.9	28.5
T5		0:100	65.2	24.6	10.2	2.6	63.6	26.6
T6	40:60	100:0	66.7	22.9	10.3	2.9	70.9	27.8
T7		75:25	61.9	27.2	11.9	2.2	78.5	25.7
T8		50:50	56.9	31.4	11.7	1.8	73.9	21.6
T9		25:75	56.2	30.9	12.8	1.8	73.1	21.9
T10		0: 100	55.1	31.5	13.4	1.7	71.2	21.5
SEM			0.04	0.06	0.26	0.05	1.05	0.14
Comparison								
R:C ratio			0.001	0.001	0.08	0.001	0.004	0.001
SBM:CM			0.001	0.009	0.81	0.004	0.54	0.004
R:C ratio × SBM:CM			0.26	0.65	0.14	0.81	0.66	0.07

R:C, ratio of roughage to concentrate at 60:40 and 40:60; SBM:CM, soybean meal replacement by cricket meal in a concentrate ratio at 100:0, 75:25, 50:50, 25:75, and 0:100, respectively; VFAs, volatile fatty acids; CH_4_, methane production calculated according to Moss et al. [24]. CH_4_ = 0.45 (C_2_) – 0.275 (C_3_) + 0.4 (C_4_); SEM, standard error of the mean.

**Table 4 insects-13-00129-t004:** Effect of cricket meal (*Gryllus bimaculatus*) (CM) in in vitro on rumen pH, ammonia–nitrogen (NH_3_-N) concentration, and protozoal population.

Treatments	R:C Ratio	SBM:CM	pH	NH_3_-N (mg/dL)	Protozoa (×10^5^ Cells/mL)
T1	60:40	100:0	6.5	18.7	9.1
T2		75:25	6.5	22.4	9.1
T3		50:50	6.5	26.3	8.7
T4		25:75	6.6	25.0	7.4
T5		0:100	6.7	26.0	8.1
T6	40:60	100:0	6.5	23.5	6.5
T7		75:25	6.4	26.6	6.1
T8		50:50	6.4	27.8	6.4
T9		25:75	6.4	28.1	7.2
T10		0:100	6.4	28.4	6.7
SEM			0.14	1.28	1.42
Comparison					
R:C ratio			0.001	0.04	0.001
SBM:CM			0.72	0.03	0.02
R:C ratio × SBM:CM			0.36	0.86	0.65

R:C, ratio of roughage to concentrate at 60:40 and 40:60; SBM:CM, soybean meal replacement by cricket meal in a concentrate ratio at 100:0, 75:25, 50:50, 25:75 and 0:100, respectively; NH_3_-N, ammonia–nitrogen concentration (mg/100 mL); SEM, standard error of the mean.

## Data Availability

Not applicable.

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
