# Peer review of "Cricket Meal (Gryllus bimaculatus) as a Protein Supplement on In Vitro Fermentation Characteristics and Methane Mitigation"

_insects, 2022, doi:10.3390/insects13020129_

Round 1
Reviewer 1 Report
Comments to the author(s): Insects-1459864 “Cricket meal (Gryllus bimaculatus) as a protein supplement on 2 in vitro fermentation characteristics and methane mitigation.”
Overall comment:
This manuscript reports on a populartopic and presents scientifically valuable results. I disagree with the authors regarding the practical importance of this study as it is unlikely that anyone would ever start feeding animal proteins (insects are animals) to ruminants in developed countries. However, the experiments in this manuscript are scientifically sound and would be valuableaddition to knowledge in this field.
Unfortunately, the manuscript is written in poor English and the results are insufficiently discussed. As this manuscript requires extensive language revision, I have refrained myself from commenting too much on language mistakes and typos in my specific comments, as it’s is not the job of a volunteer reviewer to do language revision.
Overall, I consider the science in this manuscript to be valuable and may be considered for publication after some major revisions.
General comments:
1. Language review –this manuscript requires proper English language revision. Many sentences are difficult to understand or even left unfinished. For instance, the sentence onlines 156-158 is missing an ending.
And the entire discussion section must be re-written and revised by language editor. The entire section feels like some drunk person has mumbled out random sentences that do not connect, are hard to read and contain several critical mistakes which present completely wrong meaning. A clear example is a sentence on lines 276-278, as fat does not coat the surface of microbial cells, but fibre cells. I’m certain the Authors know that, but it’s just a critical typo.
2. Simple summary – please remove all abbreviations (CP, CF, TDN and so on) from the simple summary, since you only use these once and will define these again in the rest of the manuscript.
3. The discussion is very superficial and does not mention many findings that authors have presented in the results, such as improved DM degradability (positive or negative), possible reasons why NH3-N concentration was increased so much, how does microbial adaptation possibly affect the degradation of insect proteins, possible benefits of higher EE content for the animal and post-ruminal nutrient availability.
Why did the authors used mixed diets instead of just SBM and CM in the in vitroand what were the benefits of using mixed diets. It was correctly conducted, but it remains unclear in the discussion.
Specific comments:
Line 34 change “(C2)” to “(C3)”
35-36 change “improved” to “reduced” or “decreased”. Not all readers understand what improved indicates
36 delete “up to 100%”.
37 delete “remarkably”. None of the results presented in this manuscript are remarkable.
38 delete “for SBM”, or replace it with “in concentrate mixture”
39 change “Under” to “In”
38-40 the whole sentence just repeats what was written in the three sentences before. Replace it with something more meaningful, like that CM has potential to improve rumen fermentation, or that it had no negative effects,
41 change “should be further investigated using” to “are required to further investigate the use of”
46-49 the whole sentence requires language revision
49-51 combine these two sentences. Maybe something like this: “Greenhouse gas emissions from livestock farming are high and largely caused by methane emission emanating fromrumen fermentation”.
51 delete “However”
51-52 Explain what kind of changes are proposed?
56-57/58-59 These two sentences have exactly the same meaning are. Delete the repetition and combine the meaning into one sentence.
57-58 This sentence is a controversial one. If insects are a “primary” food source and a source of high-quality protein, then why do you want to waste this resource for animal feed???Explain that in the text, why should anyone even consider using crickets as ruminant feed, instead of using these directly as human food.
69-70 That sentence is correct, but you provide no reference and that sentence is irrelevant for this current study. Delete it or provide reference.
71-74 this sentence requires language revision
73 add “polyunsaturated fatty acids” in front of PUFA, it’s the first time you use this abbreviation.
71-76 BSFL is a completely different species than crickets, and so are broilers and fish different from ruminants, so how are these sentences relevant to this study? Replace it with some relevant information about insects’ fat and/or lipids effect on rumen metabolism.
76-77 Move this sentence on the line 59, before the sentence “There are several ...”. This is good justification for your work.
80 there is a double space before “Moreover ...”
94 how many in vitroruns were used and how many replicates of each feed was used in the in vitro fermentation? Please add this information into the methods.
96 change “cricket” to “cricket meal”
97 include the make and model of the hammer mill
97 delete “in”
97 change “CM samples” to “concentrate mixtures”, cause you also used SBM not just CM
98 why did you weighed the samples into the bottles for chemical analysis? I doubt that you did this, but that’s what you have written in this sentence “... were weighed into bottles and used for chemical analysis ...”
98 change “in vitro” to “in vitro”and check this in the rest of the manuscript also
103 specify the dairy breed used for the study
103 delete “Initially”
105 change “ad libitum” to “ad libitum”
110 What software were used for modelling the gas production?
115-117 the sentence requires language revision
118 delete “kept and”
124 add “was” between “abroach ... used”
128-132 what software was used for the statistical analysis? And if there were more than one in vitro run, was that taken into account as random variable?
136-138 Please rewrite the sentence. I assume you want to say that the CP concentration was similar among treatments, but this sentence is badly written.
139 delete “Similarly”. There is nothing similar between rice straw and CP.
140 delete “respectively”. This word is incorrectly used. Please don’t use such words if you don’t know what it means or what it refers to.
Table 1 Where is the data on SBM? Please add the chemical composition of SBM also in the table.
147-153 Long and confusing sentence, and partially wrong also. SBM replacement with CM did not have effect on fraction b, but that is stated in this sentence. Make the sentence shorter so that you can focus on less factors at once.
152 Please explain what does “SBM at 100% DM” means? Does it mean that only the 100% replacement had an effect, if so then specify that in the text?
154 change “remarkably enhanced”, as I explained before, there is nothing remarkable in these results.
156 remove space “R: C”
156-158 This feels like it’s only half a sentence, without an ending. Also it presents exactly the same statement as the previous section on lines 148-152
160-162 this sentence belongs to the discussion
Table 2 how many replicates did you have in your study? Cause it’s hard to believe that you get significance below 0.01 for parameter c for instance. With all the results being almost the same and relatively high SEM, the Pvalues are strange.
171 delete “As a result”
172 Why do you write “however” when the results are similar and not different. Do you meant to write “also” instead of “however”?
173 delete “respectively” as there is no other data to be respective of in this sentence.
174-175 rewrite the sentence, it’s unclear what you want to say. Do you mean to say that there was an interaction (cause there was non according to Table 3)?
176-179 Please explain this sentence better! Do you mean to say that decreasing roughage concentrate ratio increased DM degradability and VFA andC3 proportions and decreased C2, ad CH4? If so then write it like that. However, this is what you already wrote in the lines 171-174. Do not repeat your results, there are lot moreinterestingresults you can presentfrom your tables.
179-181 This sentence is discussion again and should be moved to discussion section.
Figure 1 Many double spaces in the figure title. Also delete “as a protein source”, it’s irrelevant here.
210 add “in in vitro”
211-212 Combine these two sentences. For an example “The pH of all treatments varied between 6.4 and 6.7, being lower(P< 0.001)for substates with 40:60 R:C ratio compared to 60:40 R:C.”
212 delete “similarly” cause it’s not similar to pH, as in this case you also have effect from CM, but there was no effect of CM on pH.
213 Replace “significant shift” with an actual direction of change, like decreased or increased. This is not a murder novel, there is no need to build suspense.
214-215 what??? I have read his sentence now 15 times at least and I can’t figure out what is it that you mean to say. Please explain and rewrite this sentence
213&218 Again the “SBM at 100%DM”. explain what it means. Cause the replacement of SBM with CM has effect on all levels not only 100% replacement, so why do you only focus on the total replacement.
216 “remarkably”!!!
217 “... was additionally reduced (p < 0.05) in all supplementation groups with CM replacing 217 SBM ...” is incorrect. In Table 4, it’s shown that the protozoa numbers only reduced in the 60:40 group, but instead increased in the 40:60 group.
Table 4 add “in vitro” between “on ... rumen”. Cause you are not using in vivo measurements. Also add the time reference at which time the NH3-H and protozoa was measured. It makes the table clearer to read.
254 Change “long harsh dry season” to “experimental period”
254-255 delete the whole sentence, it was already presented in results section
255-257 what is the purpose of this sentence? Is there something wrong with rice straw? If so, explain the main problems with it.
259-260 change the sentence to “Livestock diets can be supplemented with insects, to increase the protein concentration.”
260-262 the whole sentence needs revision
262-265 that is incorrect. Your findings are very different from that of Jayanegara et al (2017). Look at the numbers that you have written, how can you say they are similar? In addition, if you read the paper Jayanegara et al (2017), you see that their results are implausible. Even if you take into account the fibre bound nitrogen, and the contribution of chitin to fibre, they still present over 100% total nutrients (CP+EE+NDF+Ash) in DM. That’s similar to election results in Soviet Union with over 100% of people taking part of the elections sometimes. When you refer to other literature,be sure to check if their results are really trustworthy and apply to you research.
265-267 Please change the whole sentence, because of course EE or fats are present in insects, that’s obvious. Instead write something like this: “According to Sánchez-Muros et al. [6], ether extract (EE) in insects are more commonly present in the form of PUFA’s”
267-271 The whole section about fatty acids in crickets is interesting, but how does it relate to the current study or to your results?
272-279 Partially correct, but why don’t you discuss why it is like that? Why is GP lower and DM digestibility lower with higher fibre content? And why is it important?
279-277 I assume it’s a typo, but fat’s cover the surface of fibre cells, not the microbial cells and that wayprevent the degradation of fibre by microbes.
280-281 It is correct that lower digestibility in insect meals results in lower h2 and therefore CH4 emissions, but why is it also a bad thing? More importantly, in your own study here you did not find reduced digestibility with CM, meaning that CM will not result to reduced nutrient availability for the animal. As you didn’t measure the post-ruminal digestibility, it’s hard to say if the similar DM degradation is a good or a bad thing.
284 what do you mean by “linearly”? I assume you mean that increasing fat content in the diet reduced CH4 emissions? If so, then explain it clearly in the text.
293-295 again, what? This sentence makes no sense! Please correct it.
296 delete “who”
295-301 Can you be sure of your statements about protein? Are sure that the slightly higher CP content in the CM mixtures 14.4% compared to 14% in SBM mixtures was enough to cause the significant increase in NH3-N? Have you calculated the potential increase of NH3-N in vitrofrom 0.4% of increased CP content?What about the protein quality and availability? Rumen microbes prefer soluble proteins to NH3-N for microbial synthesis and therefore would not degrade these proteins and non-protein nitrogen to ammonia-N. What about energy availability? Rumen microbes require energy for microbial synthesis and therefore protein breakdown and ammonia-N uptake. In your study he in vitro procedures were done in a good way as you used mixed diets, but is it possible that there was lack of energy for microbes to uptake ammonia-N? discuss this in the text. ... for example, Jayanegaraet al, 2017, incubated protein feeds as single substrates, and therefore it’s likely that he had limited energy available for microbial synthesis after12h.
311 change “in vivo” to “in vivo”
Author Response
Dear Editor;
Firstly, we wish to express our sincere thanks to you and all the reviewers for their constructive and useful comments and suggestions.
Please find details which have been closely addressed following the suggestions and recommendation of the reviewers;
Response to Reviewer 1 Comments
Specific comments:
Line 34 change “(C2)” to “(C3)”
Response: Yes, it was changed, please see in the text.
35-36 change “improved” to “reduced” or “decreased”. Not all readers understand what improved indicates
Response: Yes, we have done it, please see in the text.
36 delete “up to 100%”.
Response: Yes, it was deleted, please see in the text.
37 delete “remarkably”. None of the results presented in this manuscript are remarkable.
Response: Yes, it was deleted, please see in the text.
38 delete “for SBM”, or replace it with “in concentrate mixture”
Response: Yes, it was deleted, please see in the text.
39 change “Under” to “In”
Response: Yes, we accepted as changed, please see in the text.
38-40 the whole sentence just repeats what was written in the three sentences before. Replace it with something more meaningful, like that CM has potential to improve rumen fermentation, or that it had no negative effects,
Response: Yes, it was revised, please see in the text.
41 change “should be further investigated using” to “are required to further investigate the use of”
Response:Yes, we have done it, please see in the text.
46-49 the whole sentence requires language revision
Response: We have done revised, please see in the text.
49-51 combine these two sentences. Maybe something like this: “Greenhouse gas emissions from livestock farming are high and largely caused by methane emission emanating from rumen fermentation”.
Response: Thanks, already done, please see in the text.
51 delete “However”
Response: Yes, it was deleted, please see in the text.
51-52 Explain what kind of changes are proposed?
Response: Yes, we have revised as shown in the text.
56-57/58-59 These two sentences have exactly the same meaning are. Delete the repetition and combine the meaning into one sentence.
Response: Yes, we have edited all in the text.
57-58 This sentence is a controversial one. If insects are a “primary” food source and a source of high-quality protein, then why do you want to waste this resource for animal feed???Explain that in the text, why should anyone even consider using crickets as ruminant feed, instead of using these directly as human food.
Response: Yes, we have done it, please see in the text.
69-70 That sentence is correct, but you provide no reference and that sentence is irrelevant for this current study. Delete it or provide reference.
Response: Yes, we have provided it.
71-74 this sentence requires language revision
Response: Yes, we accepted as suggested.
73 add “polyunsaturated fatty acids” in front of PUFA, it’s the first time you use this abbreviation.
Response: Yes, we already added in the text.
71-76 BSFL is a completely different species than crickets, and so are broilers and fish different from ruminants, so how are these sentences relevant to this study? Replace it with some relevant information about insects’ fat and/or lipids effect on rumen metabolism.
73-79
Response: Yes, we have provided it.
Response: Thanks, we have already added more information,
76-77 Move this sentence on the line 59, before the sentence “There are several ...”. This is good justification for your work.
Response: Thanks, we have done revised.
80 there is a double space before “Moreover ...”
Response: Thanks, we have done it.
94 how many in vitro runs were used and how many replicates of each feed was used in the in vitro fermentation? Please add this information into the methods.
Response: We have already added more information, please see in the text.
- Each treatment contained three runs and three replications (10 treatments x 3 replications + 3 bottles of blank).
96 change “cricket” to “cricket meal”
Response: We accepted as changed.
97 include the make and model of the hammer mill
Response: Thanks, we have done it.
Rice straw, concentrate and cricket meal were dried by hot air oven at 60 °C, then ground to the length of 1 –mm (Cyclotech Mill, Tecator, Sweden).
97 delete “in”
Response: Yes, it was deleted.
97 change “CM samples” to “concentrate mixtures”, cause you also used SBM not just CM
Response: We accepted as changed.
98 why did you weighed the samples into the bottles for chemical analysis? I doubt that you did this, but that’s what you have written in this sentence “... were weighed into bottles and used for chemical analysis ...”
Response: Yes, it was revised.
98 change “in vitro” to “in vitro”and check this in the rest of the manuscript also
Response: Thanks, already done, please see in the text.
103 specify the dairy breed used for the study
Response: Yes, we already added in the text.
103 delete “Initially”
Response: Yes, it was deleted.
105 change “ad libitum” to “ad libitum”
Response: We accepted as changed.
110 What software were used for modelling the gas production?
Response: Thanks, we have done it.
Cumulative gas production data were fitted to the model of Ørskov and McDonald (1979) by Fit Curve Fitting Software
115-117 the sentence requires language revision
Response: Yes, it was revised, please see in the text.
118 delete “kept and”
Response: Yes, it was deleted.
124 add “was” between “abroach ... used”
Response: Thanks, already done, please see in the text.
128-132 what software was used for the statistical analysis? And if there were more than one in vitro run, was that taken into account as random variable?
Response: Yes, we have done it.
136-138 Please rewrite the sentence. I assume you want to say that the CP concentration was similar among treatments, but this sentence is badly written.
Response: Thanks, it was revised.
139 delete “Similarly”. There is nothing similar between rice straw and CP.
Response: Yes, it was deleted.
140 delete “respectively”. This word is incorrectly used. Please don’t use such words if you don’t know what it means or what it refers to.
Response: Yes, it was deleted.
Table 1 Where is the data on SBM? Please add the chemical composition of SBM also in the table.
Response: Thanks, we have provided it.
147-153 Long and confusing sentence, and partially wrong also. SBM replacement with CM did not have effect on fraction b, but that is stated in this sentence. Make the sentence shorter so that you can focus on less factors at once.
Response: Thanks, it was revised.
152 Please explain what does “SBM at 100% DM” means? Does it mean that only the 100% replacement had an effect, if so then specify that in the text?
Response: Yes, we have done it, please see in text.
154 change “remarkably enhanced”, as I explained before, there is nothing remarkable in these results.
Response: Thanks, it was changed.
156 remove space “R: C”
Response: Yes, we have done it.
156-158 This feels like it’s only half a sentence, without an ending. Also it presents exactly the same statement as the previous section on lines 148-152
Response: Yes, it was revised.
160-162 this sentence belongs to the discussion
Response: Yes, it was revised, please see in text.
Table 2 how many replicates did you have in your study? Cause it’s hard to believe that you get significance below 0.01 for parameter c for instance. With all the results being almost the same and relatively high SEM, the Pvalues are strange.
Response: Yes, we have done it.
Three groups of experimental bottles were established: Group 1 had gas kinetics and gas production measurement, and 3 bottles per treatment (10 treatments + 3 bottles of blank) were used. Group 2 had the pH, ruminal NH3-N, volatile fatty acids, and microbial count all examined in the same bottle. Group 3; samples were obtained at 12 h and 24 h after incubation from three bottle replicates.
171 delete “As a result”
Response: Yes, it was deleted.
172 Why do you write “however” when the results are similar and not different. Do you meant to write “also” instead of “however”?
Response: We have done edited.
173 delete “respectively” as there is no other data to be respective of in this sentence.
Response: Thanks, it was deleted, please see in the text.
174-175 rewrite the sentence, it’s unclear what you want to say. Do you mean to say that there was an interaction (cause there was non according to Table 3)?
Response: We have done revised, please see in the text.
176-179 Please explain this sentence better! Do you mean to say that decreasing roughage concentrate ratio increased DM degradability and VFA andC3 proportions and decreased C2, ad CH4? If so then write it like that. However, this is what you already wrote in the lines 171-174. Do not repeat your results, there are lot more interesting results you can present from your tables.
Response: Yes, we have revised as shown in the text.
179-181 This sentence is discussion again and should be moved to discussion section.
Response: Yes, we have done it.
Figure 1 Many double spaces in the figure title. Also delete “as a protein source”, it’s irrelevant here.
Response: Thanks, it was deleted, please see in the text.
210 add “in in vitro”
Response: We have already added.
211-212 Combine these two sentences. For an example “The pH of all treatments varied between 6.4 and 6.7, being lower (P< 0.001 ) for substates with 40:60 R:C ratio compared to 60:40 R:C.”
Response: Yes, we have done it.
212 delete “similarly” cause it’s not similar to pH, as in this case you also have effect from CM, but there was no effect of CM on pH.
Response: Thanks, it was deleted, please see in the text.
213 Replace “significant shift” with an actual direction of change, like decreased or increased. This is not a murder novel, there is no need to build suspense.
Response: Yes, we have revised as shown in the text.
214-215 what??? I have read his sentence now 15 times at least and I can’t figure out what is it that you mean to say. Please explain and rewrite this sentence
Response: Yes, we have revised as shown in the text.
213&218 Again the “SBM at 100%DM”. explain what it means. Cause the replacement of SBM with CM has effect on all levels not only 100% replacement, so why do you only focus on the total replacement.
Response: We have done edited.
216 “remarkably”!!!
Response: Thanks, it was deleted, please see in the text.
217 “... was additionally reduced (p < 0.05) in all supplementation groups with CM replacing 217 SBM ...” is incorrect. In Table 4, it’s shown that the protozoa numbers only reduced in the 60:40 group, but instead increased in the 40:60 group.
Response: Yes, we have revised as shown in the text.
Table 4 add “in vitro” between “on ... rumen”. Cause you are not using in vivo measurements. Also add the time reference at which time the NH3-H and protozoa was measured. It makes the table clearer to read.
Response: We have already added more information, please see in the text.
254 Change “long harsh dry season” to “experimental period”
Response: Thanks, it was changed.
254-255 delete the whole sentence, it was already presented in results section
Response: Thanks, it was deleted, please see in the text.
255-257 what is the purpose of this sentence? Is there something wrong with rice straw? If so, explain the main problems with it.
Response: Yes, we have revised as shown in the text.
259-260 change the sentence to “Livestock diets can be supplemented with insects, to increase the protein concentration.”
Response: Thanks, it was changed.
260-262 the whole sentence needs revision
Response: Yes, we have revised as shown in the text.
262-265 that is incorrect. Your findings are very different from that of Jayanegara et al (2017). Look at the numbers that you have written, how can you say they are similar? In addition, if you read the paper Jayanegara et al (2017), you see that their results are implausible. Even if you take into account the fibre bound nitrogen, and the contribution of chitin to fibre, they still present over 100% total nutrients (CP+EE+NDF+Ash) in DM. That’s similar to election results in Soviet Union with over 100% of people taking part of the elections sometimes. When you refer to other literature, be sure to check if their results are really trustworthy and apply to you research.
Response: Thanks, we have revised, as shown in the text.
265-267 Please change the whole sentence, because of course EE or fats are present in insects, that’s obvious. Instead write something like this: “According to Sánchez-Muros et al. [6], ether extract (EE) in insects are more commonly present in the form of PUFA’s”
Response: Thanks, it was changed, please see in the text.
267-271 The whole section about fatty acids in crickets is interesting, but how does it relate to the current study or to your results?
Response: Thanks, we have revised, as shown in the text.
Excitingly, the inclusion of oil from insects in the diet reduced the production of me-thane. Lipids might well be able to reduce enteric CH4 levels by reducing the fermentation process.
272-279 Partially correct, but why don’t you discuss why it is like that? Why is GP lower and DM digestibility lower with higher fibre content? And why is it important?
Response: Thanks, we have revised, as shown in the text.
279-277 I assume it’s a typo, but fat’s cover the surface of fibre cells, not the microbial cells and that way prevent the degradation of fibre by microbes.
Response: We have done edited, please see in the text.
280-281 It is correct that lower digestibility in insect meals results in lower h2 and therefore CH4 emissions, but why is it also a bad thing? More importantly, in your own study here you did not find reduced digestibility with CM, meaning that CM will not result to reduced nutrient availability for the animal. As you didn’t measure the post-ruminal digestibility, it’s hard to say if the similar DM degradation is a good or a bad thing.
Response: Thanks, we have revised, as shown in the text.
284 what do you mean by “linearly”? I assume you mean that increasing fat content in the diet reduced CH4 emissions? If so, then explain it clearly in the text.
Response: Thanks, we have revised, as shown in the text.
293-295 again, what? This sentence makes no sense! Please correct it.
Response: Thanks, it was revised.
296 delete “who”
Response: Yes, it was deleted.
295-301 Can you be sure of your statements about protein? Are sure that the slightly higher CP content in the CM mixtures 14.4% compared to 14% in SBM mixtures was enough to cause the significant increase in NH3-N? Have you calculated the potential increase of NH3-N in vitro from 0.4% of increased CP content? What about the protein quality and availability? Rumen microbes prefer soluble proteins to NH3-N for microbial synthesis and therefore would not degrade these proteins and non-protein nitrogen to ammonia-N. What about energy availability? Rumen microbes require energy for microbial synthesis and therefore protein breakdown and ammonia-N uptake. In your study he in vitro procedures were done in a good way as you used mixed diets, but is it possible that there was lack of energy for microbes to uptake ammonia-N? discuss this in the text. ... for example, Jayanegaraet al, 2017, incubated protein feeds as single substrates, and therefore it’s likely that he had limited energy available for microbial synthesis after12h.
Response: Thanks, we have revised, as shown in the text.
311 change “in vivo” to “in vivo”
Response: Thanks, it was changed, please see in the text.
Finally, we greatly appreciate all your useful comments and suggestions which has tremendously improved the manuscript quality and should be ready for publication.
Prof. Dr. Metha Wanapat
On behalf of the authors

Reviewer 2 Report
Dear Editor,
Thank you for the opportunity to review the manuscript entitled: "Cricket meal (Gryllus bimaculatus) as a protein supplement on in vitro fermentation characteristics and methane mitigation"
The subject is interesting given the rising costs of soy protein products in animal feeds. However, major corrections are needed to improve the scientific soundness of the journal. The introduction is rather chaotic, it should be arranged to provide the background, problem statement, justification, objective and hypothesis without actually adding these as headings. There is some element of repetition as well, which need to be addressed. The result section also needs to show how the means were compared by adding superscripts on the means to denote significant differences. A synthetic conclusion is needed. Also see comments below:
Line 16/38: Those are not countries. The statement is also incorrect because not all insects are human-edible.
Line 25: Remove the scientific term it is already given in line 19/20
Line 28: The word 'meal' should come after the scientific term of the crickets
Line 30 and 91: Don't capitalize the 'C' in completely randomised design..
Line 33: Name the kinetics
Line 35: Proprionate ans acetate are given the same acronyms. C2:C3 ration should be defined at first mention including all other acronyms/abbreviations
Line 37 and 154: Remove the word remarkably, it doesn't denote statistical difference. Therefore, it is statistically incorrect.
Line 39: modify the sentence with 'by enhancing'
Line 59 - 61: You need more than one reference there. See:
Marareni, M. & Mnisi C.M., 2020. Growth performance, serum biochemistry and meat quality traits of Jumbo quails fed with mopane worm (Imbrasia belina) meal-containing diets. Veterinary and Animal Science 10, 100141.
Line 118: Italicize the 'g' so that it doesn't represent grams.
Line 120: Define the VFAs even there
Line 127: Elaborate a little bit, a formula can be provided.
Italicise 'in vitro', 'ad libitum' and all other scientific tems.
Thanks
Author Response
Dear Editor;
Firstly, we wish to express our sincere thanks to you and all the reviewers for their constructive and useful comments and suggestions.
Please find details which have been closely addressed following the suggestions and recommendation of the reviewers;
Response to Reviewer 2 Comments
Line 16/38: Those are not countries. The statement is also incorrect because not all insects are human-edible.
Response: Thanks, we have done edited, please see in the text.
Line 25: Remove the scientific term it is already given in line 19/20
Response: Yes, we have done it, please see in the text.
Line 28: The word 'meal' should come after the scientific term of the crickets
Response: Yes, we have provided it.
Line 30 and 91: Don't capitalize the 'C' in completely randomised design..
Response: Thanks, we have done edited.
Line 33: Name the kinetics
Response: Thanks, we have done it.
Line 35: Proprionate and acetate are given the same acronyms. C2:C3 ration should be defined at first mention including all other acronyms/abbreviations
Response: Thanks, already done, please see in the text.
Line 37 and 154: Remove the word remarkably, it doesn't denote statistical difference. Therefore, it is statistically incorrect.
Response: Yes, we have done it, please see in the text.
Line 39: modify the sentence with 'by enhancing'
Response: Thanks, it was revised.
Line 59 - 61: You need more than one reference there. See:
Marareni, M. & Mnisi C.M., 2020. Growth performance, serum biochemistry and meat quality traits of Jumbo quails fed with mopane worm (Imbrasia belina) meal-containing diets. Veterinary and Animal Science 10, 100141.
Response: We have already added more information, please see in the text.
Line 118: Italicize the 'g' so that it doesn't represent grams.
Response: Thanks, we have done it.
Line 120: Define the VFAs even there
Response: Yes, we have edited it.
Line 127: Elaborate a little bit, a formula can be provided.
Response: Yes, we have done it, please see in the text.
Italicise 'in vitro', 'ad libitum' and all other scientific tems.
Response: Thanks, already done.
Finally, we greatly appreciate all your useful comments and suggestions which has tremendously improved the manuscript quality and should be ready for publication.
Prof. Dr. Metha Wanapat
On behalf of the authors

Reviewer 3 Report
insects-1459864
In the paper entitled “Cricket meal (Gryllus bimaculatus) as a protein supplement on 2 in vitro fermentation characteristics and methane mitigation” the effect of the Gryllus bimaculatus as a protein source was investigated in in vitro gas production technique.
Abstract:
Line 30: add “in ruminants”.
Introduction:
Line 67: please add information concerning the role of chitin in feed.
Mat & Met:
Line 95: please add information concerning the cricket flour origin.
Please add information concerning the methane, NH3 and VFA analyses.
Why do you have chosen 96 h of incubation? Usually, when the complete diet are incubated, it’s commonly used 120 h of incubation.
Discussion:
Please improve the discussion, you can use these papers:
Boussaada et al. (2018) Annals of Animal Science, 18(3):753-767. (protozoa)
Benchaar et al. (2008). Dairy Sci., 91: 4765–4777. (protozoa)
Jayanegara et al. 10.3923/pjbs.2017.523.529 (chitin content)
Conclusions:
Please remove lines 310-312, the conclusions are reported later.
Author Response
Dear Editor;
Firstly, we wish to express our sincere thanks to you and all the reviewers for their constructive and useful comments and suggestions.
Please find details which have been closely addressed following the suggestions and recommendation of the reviewers;
Response to Reviewer 3 Comments
Abstract:
Line 30: add “in ruminants”.
Response: Thanks, already done.
Introduction:
Line 67: please add information concerning the role of chitin in feed.
Response: We have already added more information, please see in the text.
Mat & Met:
Line 95: please add information concerning the cricket flour origin.
Response: We have already added more information, please see in the text.
The cricket farming and the cricket flour production from Khon kaen, Thailand.
Please add information concerning the methane, NH3 and VFA analyses.
Response: We have already added more information, please see in the text.
Why do you have chosen 96 h of incubation? Usually, when the complete diet are incubated, it’s commonly used 120 h of incubation.
Response: Yes, we were aware of your kind comment, unfortunately, we failed to continue, hopefully in the next experiment, we will do it, thanks.
Discussion:
Please improve the discussion, you can use these papers:
Boussaada et al. (2018) Annals of Animal Science, 18(3):753-767. (protozoa)
Benchaar et al. (2008). Dairy Sci., 91: 4765–4777. (protozoa)
Jayanegara et al. 10.3923/pjbs.2017.523.529 (chitin content)
Response: Thanks, we have already added more information, please see in the text.
Conclusions:
Please remove lines 310-312, the conclusions are reported later.
Response: Yes, we have edited it. Please see in the text.
Finally, we greatly appreciate all your useful comments and suggestions which has tremendously improved the manuscript quality and should be ready for publication.
Prof. Dr. Metha Wanapat
On behalf of the authors

Round 2
Reviewer 1 Report
General comments:
- Language review – this manuscript would still benefit a lot from English language review
- Discussion – the whole discussion is still badly written, with full of meaningless statements from other authors and disconnecting sentences. It’s hard to read.
Also the discussion would be better if protein degradation and ammonia N concentrations would be discussed in more detail, however, since the study focuses only on methane then I guess it’s not so important. It’s just a lost opportunity for the authors.
Specific comments:
Line
54-55 that sentence is not completely correct. Insects have been a traditional food source in these regions, but have recently become an interest as alternative protein source in other parts such as Europe and North America also.
115 italicise “in vitro”. Check throughout the manuscript for other such mistakes.
133 change “H2PO4” to “H2PO4”
234 Improved nutritive value of what? Did the rice straw improve nutritive value of a diet/ration, or did the treatment improved the nutritive value of rice straw? It’s not clear what is it that you want to say
239-241 same comment as last time, these findings are not similar. 38.5% and 67.7%CP, or 23.1% and 39.3%NDF are not similar. Please rewrite or delete this sentence, as it’s a false statement.
Either explain why the CP concentration of your crickets was so much lower than that of the other authors, or it would be much more important to state that the SBM and CM had similar CP and EE concentration.
241-244 Make this sentence as a separate sentence from the previous part (239-241), cause these provide different type of data.
254 delete “to”
264 delete “report by Machmuller”
264-266 specify if this sentence is from the previous reference. At the moment it looks like it’s about your own study, but you have not used different insect oils.
268-270 You state that Russell and Rychlik concluded that changes in rumen pH influence metabolic disorders. But what does that mean for your own results? Does that mean that the small changes in pH found in your study was good or bad? What is the purpose of that sentence?
Author Response
Dear Editor;
Firstly, we wish to express our sincere thanks to you and all the reviewers for their constructive and useful comments and suggestions.
Please find details which have been closely addressed following the suggestions and recommendation of the reviewers;
Response to Reviewer 1
Comments
- Language review – this manuscript would still benefit a lot from English language review
Response: Thank you for your valuable suggestions. We have revised it.
- Discussion – the whole discussion is still badly written, with full of meaningless statements from other authors and disconnecting sentences. It’s hard to read. Also the discussion would be better if protein degradation and ammonia N concentrations would be discussed in more detail, however, since the study focuses only on methane then I guess it’s not so important. It’s just a lost opportunity for the authors.
Response: Thank you for your valuable suggestions. We have revised it. Kazemi-Bonchenari et al. [47] who demonstrated that higher amount of protein in the diet leads to higher concentrations of ruminal NH3-N in the rumen. Proteins in the rumen are broken down into ammonia by proteolytic microbes [48]. Jayanegara et al. [49] showed that the concentration of rumen degradable protein in concentrate was higher than in forage, thereby contributes to higher ammonia production. In the present study, higher levels of concentrate diet and levels of CM used to replace SBM enhanced ruminal NH3-N concentration. This is most likely owing to the high protein content of CM. It could also be attributed to the ability of ruminal bacteria to produce stimulatory substances and even protein, or to changes in the abundance of microbes with proteolytic activity. Furthermore, changing the R:C ratio in the diet from 60:40 to 40:60 raised the ruminal NH3-N concentration. Similarly, Phesatcha et al. [50] and Viennasay et al. [51] discovered that rumen NH3-N concentrations increased dramatically with decreasing R:C ratio.
Specific comments:
Line
54-55 that sentence is not completely correct. Insects have been a traditional food source in these regions, but have recently become an interest as alternative protein source in other parts such as Europe and North America also.
Response: Thanks, we have revised it.
115 italicise “in vitro”. Check throughout the manuscript for other such mistakes.
Response: Thanks, we have revised throughout the manuscript.
133 change “H2PO4” to “H2PO4”
Response: Thanks, we have revised it.
234 Improved nutritive value of what? Did the rice straw improve nutritive value of a diet/ration, or did the treatment improved the nutritive value of rice straw? It’s not clear what is it that you want to say
Response: Thanks, we have revised it. Rice straw was found to be abundantly available in many countries and was fed ad libitum during the experimental period. Wanapat et al. [33] described feeding rice straw with a concentrate mixture containing a high density of energy and protein could be beneficial in increasing its utilization.
239-241 same comment as last time, these findings are not similar. 38.5% and 67.7%CP, or 23.1% and 39.3%NDF are not similar. Please rewrite or delete this sentence, as it’s a false statement.
Either explain why the CP concentration of your crickets was so much lower than that of the other authors, or it would be much more important to state that the SBM and CM had similar CP and EE concentration.
Response: Thanks, we have carefully checked and revised it “Cricket meal used in this experiment contained 68.5 %CP, 33.1% NDF, 12.7% ADF, and 12.5% EE, this finding was similar to Jayanegara et al. [5] stated that cricket meal contained 67.7% CP, 39.3% NDF, 10.8% ADF and 14.5.% EE. Sánchez-Muros et al. [9] showed that insect ether extract (EE) contained polyunsaturated fatty acids (PUFAs).
241-244 Make this sentence as a separate sentence from the previous part (239-241), cause these provide different type of data.
Response: Thanks, we have revised it.
254 delete “to”
Response: Thanks, we have deleted it.
264 delete “report by Machmuller”
Response: Thanks, we have deleted it.
264-266 specify if this sentence is from the previous reference. At the moment it looks like it’s about your own study, but you have not used different insect oils.
Response: Thanks, we have revised it.
268-270 You state that Russell and Rychlik concluded that changes in rumen pH influence metabolic disorders. But what does that mean for your own results? Does that mean that the small changes in pH found in your study was good or bad? What is the purpose of that sentence?
Response: Thanks, we have revised it.

Reviewer 3 Report
The paper entitled " Cricket meal (Gryllus bimaculatus) as a protein supplement on in vitro fermentation characteristics and methane mitigation " by Phesatcha et al. has been improved according to the reviewers suggestions. All my requests have been taken into consideration. The paper could be accepted in the current form.
Author Response
Dear Reviewer;
Firstly, we wish to express our sincere thanks to you and all the reviewers for their constructive and useful comments and suggestions.
Finally, we greatly appreciate all your useful comments and suggestions which has tremendously improved the manuscript quality and should be ready for publication.
Prof. Dr. Metha Wanapat
On behalf of the authors
